# Days of Antibiotic Spectrum Coverage Trends and Assessment in Patients with Bloodstream Infections: A Japanese University Hospital Pilot Study

**DOI:** 10.3390/antibiotics11121745

**Published:** 2022-12-02

**Authors:** Masayuki Maeda, Mari Nakata, Yuika Naito, Kozue Yamaguchi, Kaho Yamada, Ryoko Kinase, Takahiro Takuma, Rintaro On, Issei Tokimatsu

**Affiliations:** 1Division of Infection Control Sciences, Department of Clinical Pharmacy, Showa University School of Pharmacy, 1-5-8, Hatanodai, Shinagawa-ku, Tokyo 142-8555, Japan; 2Department of Hospital Pharmaceutics, Showa University School of Pharmacy, 1-5-8, Hatanodai, Shinagawa-ku, Tokyo 142-8555, Japan; 3Division of Infection Diseases, Department of Medicine, Showa University School of Medicine, 1-5-8, Hatanodai, Shinagawa-ku, Tokyo 142-8555, Japan

**Keywords:** antimicrobial use metric, antimicrobial stewardship, bloodstream infection, antibiotic spectrum score, de-escalation

## Abstract

The antibiotic spectrum is not reflected in conventional antimicrobial metrics. Days of antibiotic spectrum coverage (DASC) is a novel quantitative metric for antimicrobial consumption developed with consideration of the antibiotic spectrum. However, there were no data regarding disease and pathogen-specific DASC. Thus, this study aimed to evaluate the DASC trend in patients with bloodstream infections (BSIs). DASC and days of therapy (DOT) of in-patients with positive blood culture results during a 2-year interval were evaluated. Data were aggregated to calculate the DASC, DOT, and DASC/DOT per patient stratified by pathogens. During the 2-year study period, 1443 positive blood culture cases were identified, including 265 suspected cases of contamination. The overall DASC, DASC/patient, DOT, DOT/patient, and DASC/DOT metrics were 226,626; 157.1; 28,778; 19.9; and 7.9, respectively. A strong correlation was observed between DASC and DOT, as well as DASC/patient and DOT/patient. Conversely, DASC/DOT had no correlation with other metrics. The combination of DASC and DOT would be a useful benchmark for the overuse and misuse evaluation of antimicrobial therapy in BSIs. Notably, DASC/DOT would be a robust metric to evaluate the antibiotic spectrum that was selected for patients with BSIs.

## 1. Introduction

Defined daily doses and days of therapy (DOT) are used to quantify antimicrobial consumption around the world [1,2]. These metrics have been monitored to measure the antimicrobial stewardship program (ASP) process or outcome in inpatient settings [3,4,5]. The de-escalation strategy is one of the ASPs that reduce unnecessary broad-spectrum antibiotic use, which can reduce bacterial resistance and health costs [6,7,8,9]. However, conventional antibiotic metrics did not reflect the antibiotic spectrum or targeted bacteria. Additionally, the de-escalation strategy has not been sufficiently evaluated because of its unclear definition [9,10,11,12]. Previous studies evaluated de-escalation strategy by antimicrobial stewardship interventions, although each study’s definition varied [13,14,15,16,17].

Kakiuchi et al. developed a novel quantitative metric for antimicrobial consumption with antibiotic spectrum consideration to address the problems of conventional metrics [18]. The individual antibiotic spectrum coverage (ASC) score and its dosing period are used to calculate the days of antibiotic spectrum coverage (DASC). DASC can evaluate both overall antibiotic consumption and antibiotic spectrum [18]. Therefore, we considered that DASC will be suitable for disease or pathogen-specific antimicrobial use evaluation, including a de-escalation strategy [17,19]. However, data on disease or pathogen-specific DASC remain lacking to evaluate ASPs. Thus, this study aimed to evaluate the trend of DASC in patients with bloodstream infections.

## 2. Results

### 2.1. Patient Characteristics, DASC, and DOT Metrics in This Study

A total of 1443 positive blood cultures cases were identified that included 265 suspected cases of contamination during the 2-year study period. Table 1 shows the stratified BSI cases by age category.

Table 2 shows the overall DASC and DOT metrics in this study population. Each metric’s trends were comparable between 2020 and 2021.

### 2.2. DASC, DOT, and DASC/DOT Trends in BSI Patients by Month

Figure 1 and Figure 2 show the DASC, DASC/patient, DOT, and DOT/patients trends by month. These metrics transitioned to parallel by month. In contrast, the DASC/DOT transitions were independent of DASC or DOT (Figure 3).

### 2.3. Relationship between DASC, DOT, DASC/DOT, and BSI Patient Number by Month

The relationships among DASC, DASC/patient, DOT, DOT/patient, DASC/DOT, and no. of patients per month are shown in Figure 4. A strong correlation was observed between DASC and DOT, as well as DASC/patient and DOT/patient. In contrast, DASC/DOT had no correlation with other metrics.

### 2.4. DASC/Patient, DOT/Patient, and DASC/DOT Comparisons Which Were Stratified by Pathogens

The DASC/patient, DOT/patient, and DASC/DOT distribution, which were stratified by BSI pathogens are shown in Figure 5, Figure 6 and Figure 7. DASC/patient and DOT/patient high scores were Gram-positive bacteria compared with Gram-negative bacteria (Figure 5 and Figure 6). Conversely, the DASC/DOT of Gram-positive bacteria was lower compared with Gram-negative bacteria (Figure 7).

## 3. Discussion

The present study revealed that the DASC metric could be applied to antimicrobial use evaluation in BSI patients. Previous reports on spectrum scoring have evaluated the de-escalation strategy because conventional antibiotic use metrics do not mirror the antibiotic spectrum [20,21,22,23,24]. However, the antibiotic spectrum score application was limited to the frequently used antibiotics and infectious diseases. Moreover, the calculation process of scoring was complicated [21,22]. DASC was simply based on dichotomous scores for efficacy against microorganisms [18]. Therefore, the ASC scores list was exhaustive, as well as simple to set up.

In this study, parallel changes in DASC and DOT per month were observed. A low DASC/DOT score would indicate that there was no broad-spectrum antibiotic overuse, although both DASC and DOT increased in September 2020. Conversely, no DASC and DOT increment in January 2021 with a DASC/DOT increment would be indicative of broad-spectrum antibiotics overuse. The long-term use intervention or promoting antibiotic de-escalation would be needed during the periods according to the results of these metrics. In addition, the correlation analysis showed that DASC/DOT had no correlation with other metrics. DASC/DOT was an independent and robust metric that could evaluate the antibiotic spectrum consistent with the previous study [18]. However, in the case of empirical therapy using broad-spectrum and stopping antibiotics immediately, DASC/DOT reflects a broad-spectrum antibiotics score. The results of this study suggested that a DASC and DASC/DOT combination is a metric that can evaluate broad-spectrum antibiotics overuse in BSIs.

Pathogen-specific DASCs were evaluated to provide utility ASP evaluation. The DASC and DOT of Gram-positive bacteria were higher than Gram-negative bacteria. Gram-positive bacterial infections need long-term antibiotic treatment, such as infective endocarditis. A high DASC was caused by a DOT increment, which was long-term antibiotic use. Conversely, a high DASC/DOT was observed in Gram-negative bacteria. Antibiotics that have activity against Gram-negative bacteria typically have high ASC scores [18]. The metrics stratified by BSI pathogens will clarify the object of ASP interventions. Notably, each metric suspected of case contamination in this study was evaluated. A contaminated case of blood culture is associated with unnecessary treatment [25]. ASPs should target to reduce the DASC of contaminated cases including BSIs due to fungi, although this study could not discriminate against other existing infections [26,27].

Ideally, antibiotic use metrics should evaluate antimicrobial use appropriateness or standardization [28,29,30]. The standardized antimicrobial administration ratio (SAAR), which was risk-adjusted for hospital-level and unit-level factors, has been implemented as a standardized benchmark of antibiotic use in the United States, although conventional metrics could not be quantitatively evaluated for appropriateness [28]. Pathogen-specific DASCs will potentially be applied as a standardized metric in antimicrobial use. Cefazolin has been a recommended treatment for *S. aureus* bacteremia [31,32]. The treatment duration is 14 days for uncomplicated infections and 4–8 weeks for complicated infections, including infective endocarditis [31,32,33]. An ASC score of 3 for cefazolin and its dosing period-derived standardized DASC/patient is 42 in uncomplicated infections to 84–168 in complicated infections. Moreover, carbapenems are standard antibiotics for extended-spectrum beta-lactamase (ESBL) producing Enterobacterales infections [34]. An ASC score of 12 for meropenem and a 14-day treatment period of derived standardized DASC/patient as 168 in complicated urinary tract infections [35]. Fewer than both 168 DASC/patient and 12 DASC/DOT indicate de-escalation from meropenem to narrower-spectrum antibiotics. A DASC, DOT, and DASC/DOT combination stratified by BSI pathogens will be able to evaluate standardized and appropriate antimicrobial use. However, standard therapy periods are different by infection focus and severity. Further studies are needed to determine the disease-specific DASC appropriateness.

The present study has several limitations. First, this was a single-center pilot study conducted in Tokyo, Japan, and our findings cannot be generalized to another geographical area with different treatment standards and epidemiological BSI characteristics. Multicenter and large-size studies are needed. Second, this study overlooked the correlation between the metrics and outcomes, such as AMR, *Clostridioides difficile* infection, and cost saving. Third, the actual ASPs interventions were not evaluated. The previous study suggested that the efforts of ASPs to use narrow-spectrum antibiotics through empiric therapy or de-escalation are not evaluated in the DOT-based metrics [18]. Further studies are needed to evaluate the DASC application for the process and outcome of ASP interventions. Finally, the concept of DASC has not been widespread in Japan. The development of an electronic surveillance system would be needed to aggregate the DASC data in each hospital.

In conclusion, the study revealed that the DASC could evaluate antimicrobial use in patients with BSIs. The combination of DASC and DOT would be a useful benchmark for the evaluation of antimicrobial therapy overuse and misuse in BSIs. Notably, DASC/DOT would be a robust metric to evaluate the antibiotic spectrum that was selected for patients with BSIs. Disease and pathogen-specific DASC will be a benchmarking function that will lead to promoting antimicrobial stewardship because of the quantitative antibiotic spectrum evaluation.

## 4. Materials and Methods

### 4.1. Study Setting and Design

This retrospective study was performed at the Showa University Hospital, an 815-bed tertiary teaching hospital in Japan. Inclusions were in-patients with positive blood culture results during a 2-year interval from 1 April 2020 to 31 March 2022. Patient age, antimicrobial regimen, isolated microorganisms, and AMR data were extracted from patients’ medical records.

### 4.2. Calculation of Antimicrobial Use Metrics

Antimicrobial use data were collected from the initial blood sampling day for culture. DASC was calculated by the ASC score defined in the previous study [18]. If an antimicrobial that was not defined by the ASC score was used, a similar class antibiotic score was adopted according to the method of the prior study [18]. DASC/patient indicates the total amount of antibiotic spectrum and dosing period per patient. DOT/patient indicates mainly the dosing period per patient in addition to the number of antibiotic combinations. DASC/DOT indicates the average selected ASC score per patient.

### 4.3. Microbiologic Results of Blood Culture and AMR Definitions

BSIs were defined as organism isolation from positive blood culture. A suspected case of contamination was defined as isolating skin-resident or environmental bacteria from single blood culture which was not an identified specific infection focus [36,37]. BD BACTEC FX System (Becton, Dickinson and Company, Franklin Lakes, NJ, USA) was used to perform blood cultures. MicroScan WalkAway system (Beckman Coulter, Inc., Tokyo, Japan) was used to screen isolated strains.

The isolated pathogens were divided into major bacteria that cause BSIs to evaluate pathogen-specific antimicrobial use metrics. Furthermore, the AMR was tested in five microorganisms: *Staphylococcus aureus*, *Escherichia coli*, *Klebsiella* species, *Enterobacter* species (including *K. aerogenes*), and *Pseudomonas aeruginosa* [19]. The clinical breakpoints of the Clinical and Laboratory Standards Institute (CLSI) were used to define AMR [38]. Antimicrobial-resistant *P. aeruginosa* was resistant to at least two of the following antimicrobials: fluoroquinolones, aminoglycosides, and carbapenems. Methicillin resistance in *S. aureus* was confirmed by oxacillin and cefoxitin testing. The double disk synergy test was used to determine ESBL production in *E. coli* and *Klebsiella* spp. Disks containing cephalosporins (ceftazidime and cefotaxime) were applied next to amoxicillin–clavulanic acid disks. Antimicrobial-resistant *Enterobacter* spp. was defined by its resistance to third-generation cephalosporins or carbapenems that include the isolation and identification of a minimum inhibitory concentration (MICs) of meropenem ≥2 µg/mL or MICs of imipenem ≥2 µg/mL and cefmetazole ≥64 µg/mL by the Japanese criteria [39].

### 4.4. Statistical Analysis

Spearman’s rank correlation analysis was used to calculate the coefficient (ρ) among antimicrobial use metrics each month. Statistical tests were two-tailed, and *p*-values < 0.05 were considered significant. SPSS statistics version 23.0 (IBM Japan, Tokyo, Japan) was used to perform statistical analysis.

### 4.5. Ethics

The Showa University Research Ethics Committee (Approval no. 22-173-A) approved the study.

## Figures and Tables

**Figure 1 antibiotics-11-01745-f001:**
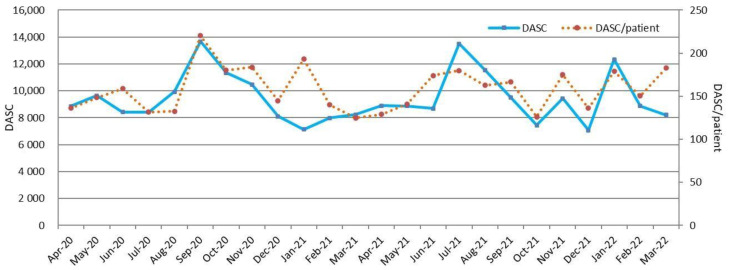
Monthly DASC and DASC/patient trends during this 2-year period. DASC, days of antibiotic spectrum coverage.

**Figure 2 antibiotics-11-01745-f002:**
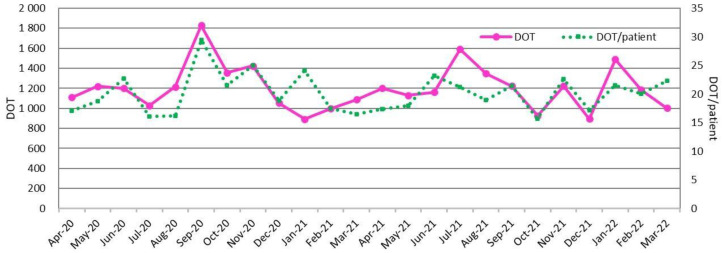
Monthly DOT and DOT/patient trends during this 2-year period. DOT, days of therapy.

**Figure 3 antibiotics-11-01745-f003:**
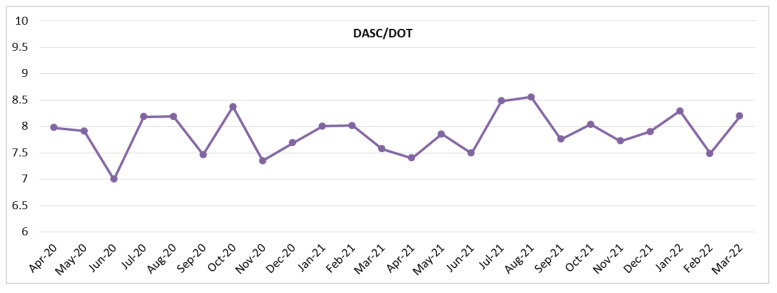
Monthly DASC/DOT trend during this 2-year period. DASC, days of antibiotic spectrum coverage; DOT, days of therapy.

**Figure 4 antibiotics-11-01745-f004:**
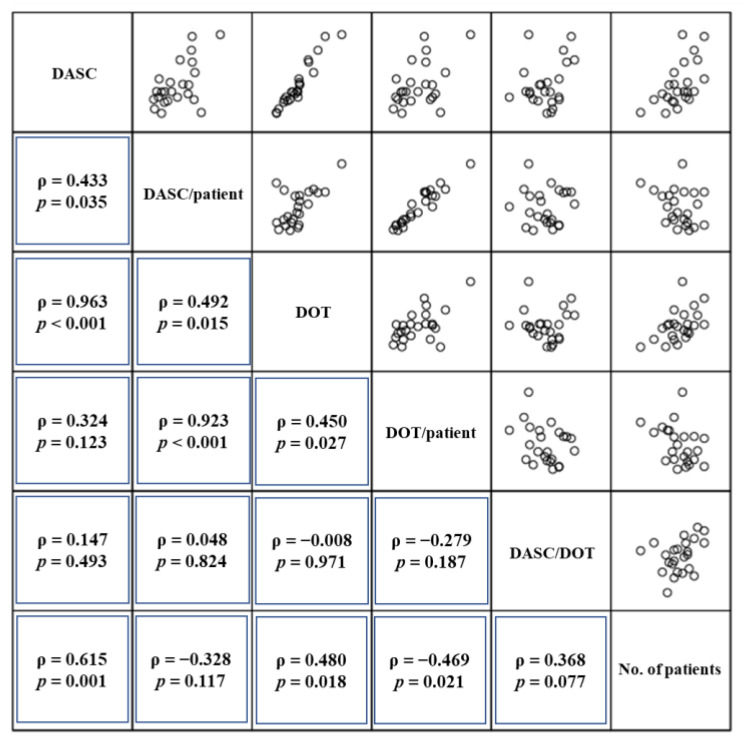
The scatterplot among DASC, DASC/patient, DOT, DOT/patient, DASC/DOT, and the number of patients per month. The *x*- and *y*-axes indicate each metric, while the plots indicate each month’s observed values. Diagonal cells show Spearman’s correlation analysis results. A strong correlation was observed between DASC and DOT (ρ = 0.963) as well as DASC/patient and DOT/patient (ρ = 0.923). In contrast, DASC/DOT had no correlation with other metrics (ρ = −0.279–0.368). DASC, days of antibiotic spectrum coverage; DOT, days of therapy.

**Figure 5 antibiotics-11-01745-f005:**
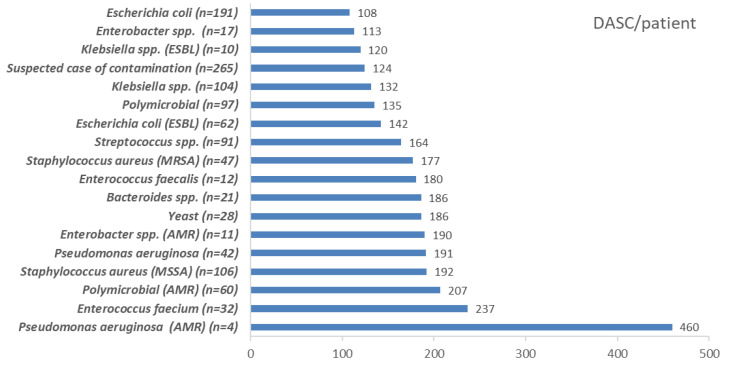
The DASC/patient distribution stratified by BSI pathogens. DASC, days of antibiotic spectrum coverage; BSI, bloodstream infection; MSSA, methicillin-susceptible *Staphylococcus aureus*; MRSA, methicillin-resistant *Staphylococcus aureus*; ESBL, extended-spectrum beta-lactamase; AMR, antimicrobial resistance.

**Figure 6 antibiotics-11-01745-f006:**
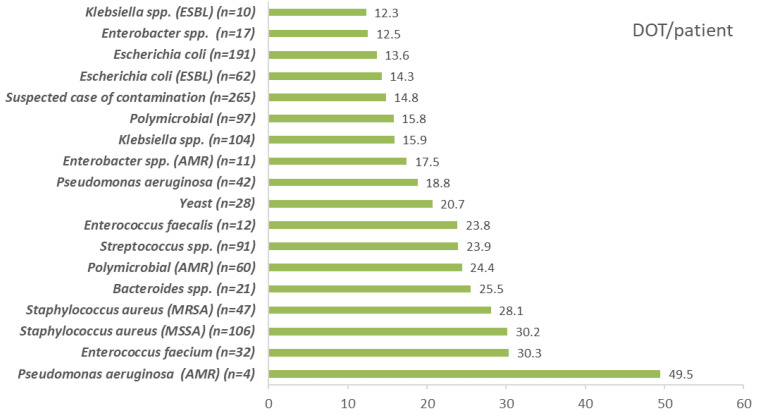
The DOT/patient distribution stratified by BSI pathogens. DOT, days of therapy; BSI, bloodstream infection; MSSA, methicillin-susceptible *Staphylococcus aureus*; MRSA, methicillin-resistant *Staphylococcus aureus*; ESBL, extended-spectrum beta-lactamase; AMR, antimicrobial resistance.

**Figure 7 antibiotics-11-01745-f007:**
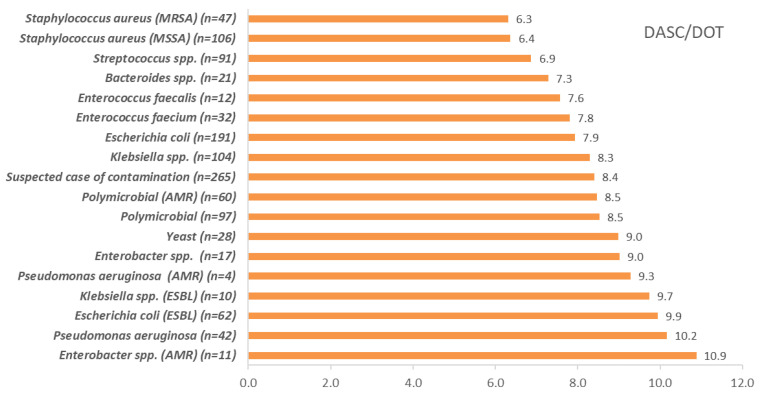
The DASC/DOT distribution stratified by BSI pathogens. DASC, days of antibiotic spectrum coverage; DOT, days of therapy; MSSA, methicillin-susceptible *Staphylococcus aureus*; MRSA, methicillin-resistant *Staphylococcus aureus*; ESBL, extended-spectrum beta-lactamase; AMR, antimicrobial resistance.

**Table 1 antibiotics-11-01745-t001:** Positive blood culture cases and cases without contamination stratified by age category.

Age Category	No. of Cases (All, *n* = 1443)	No. of Cases (Without Contamination, *n* = 1178)
0–5	46 (3.2)	35 (3.0)
6–14	9 (0.6)	8 (0.7)
15–49	121 (8.4)	93 (7.9)
50–64	232 (16.1)	189 (16.0)
65–74	324 (22.5)	264 (22.4)
≥75	711 (49.3)	589 (50.0)

Data were presented as *n* (%).

**Table 2 antibiotics-11-01745-t002:** The overall DASC and DOT metrics in this study population.

Year	DASC	DASC/Patient	DOT	DOT/Patient	DASC/DOT
2020	112,235	155.9	14,403	20.0	7.8
2021	114,391	158.2	14,375	19.9	8.0
Total	226,626	157.1	28,778	19.9	7.9

DASC, days of spectrum coverage; DOT, days of therapy.

## Data Availability

Data are available on request due to restrictions, e.g., privacy or ethical.

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
