# Peer review of "Days of Antibiotic Spectrum Coverage Trends and Assessment in Patients with Bloodstream Infections: A Japanese University Hospital Pilot Study"

_antibiotics, 2022, doi:10.3390/antibiotics11121745_

Round 1
Reviewer 1 Report
This study indicated the relationship DASC, DOT, DASC/DOT metrics for evaluation as benchmark for over- or miss-use of antimicrobials for patients of blood stream infection. This pilot study is limited data but novel and interesting work.
1. Lines 134-136: The results indicated great relationship between DASC and DOT. Therefore, metrics of the evaluation may be sufficient in DASC and DASC/DOT or DOT and DASC/DOT. Please indicate why three metrics (DASC, DOT, DASC/DOT) are needed.
2. Limitation section: As the concept of DASC is not yet widespread, it is recommended that the limitations with regard to DASC are also briefly described with reference to [19] etc.
Author Response
Reviewer #1:
Thank you very much for reviewing our manuscript together with the comments. The suggestions for improving the manuscript were very helpful in preparing a revised submission. We revised the manuscript according to your suggestions.
- Lines 134-136: The results indicated great relationship between DASC and DOT. Therefore, metrics of the evaluation may be sufficient in DASC and DASC/DOT or DOT and DASC/DOT. Please indicate why three metrics (DASC, DOT, DASC/DOT) are needed.
Response:
Thank you for the comment. As the reviewer's pointing out, a strong correlation was observed between DASC and DOT. We agree with the reviewer's comment that it is sufficient to evaluate the metrics using only DASC and DASC/DOT. We revised that sentence.
Change(s): P6 L136
- Limitation section: As the concept of DASC is not yet widespread, it is recommended that the limitations with regard to DASC are also briefly described with reference to [19] etc.
Response:
Thank you for the comment. We referred to the previous DASC study. Since this metric has just been developed, it is not widespread in the U.S. as well as in Japan where the DOTs are widely collected using an electronic surveillance system. We believe that it will be necessary to develop an electronic database for DASC collection. We have added this insight to the limitation section.
Change(s): P7 L178-180
Finally, the concept of DASC has not been widespread in Japan. The development of an electronic surveillance system would be needed to aggregate the DASC data in each hospital.

Reviewer 2 Report
This research article suggests new metrics to evaluate the antibiotic spectrum for bloodstream infection patients. The researchers suggest DASC and DOT would be useful tools for evaluating appropriate antibiotic use in BSIs.
1. Ref: It contains some inappropriate references. Some unnecessary references should be deleted. For example, I'm not sure why ref #1 is included in the reference list.
2. It would be nice to rearrange figures 5, 6, and 7. If the numbers are arranged in ascending order, it seems to be easier to interpret the data intuitively.
I felt that the manuscript story was not well organized overall. Although DASC and DOT related metrics were found to have a relationship, it is not persuasive enough that they have advantages over the previously used metrics.
Author Response
Reviewer #2:
Thank you very much for reviewing our manuscript together with the comments. The suggestions for improving the manuscript were very helpful in preparing a revised submission. We revised the manuscript according to your suggestions.
- Ref: It contains some inappropriate references. Some unnecessary references should be deleted. For example, I'm not sure why ref #1 is included in the reference list.
Response:
According to the reviewer's comment, we revised the reference list. Reference No. 1, 20, and 30 were deleted.
Change(s): Please refer each reference number and reference list in the manuscript.
- It would be nice to rearrange figures 5, 6, and 7. If the numbers are arranged in ascending order, it seems to be easier to interpret the data intuitively.
Response:
According to the reviewer’s comment, we arranged them in ascending order.
Change(s): We replaced the Fig. 5, 6, and 7 (Page 4-5).
- I felt that the manuscript story was not well organized overall. Although DASC and DOT related metrics were found to have a relationship, it is not persuasive enough that they have advantages over the previously used metrics.
Response:
Thank you for the comment. We could not evaluate DASC utility in actual activities compared with conventional metrics. However, the previous study indicated the utility and advantage of DASC compared with other metrics. We have added this insight to the limitation section.
Change(s): P6-7 L174-177
The previous study suggested that the efforts of ASPs to use narrower-spectrum antibiotics through empiric therapy or de-escalation are not evaluated in the DOT-based metrics.

Round 2
Reviewer 2 Report
Thank you for taking my suggestion into account and for making the corrections. I have no further comments.